# Balancing Page Endurance Variation Between Layers to Extend 3D NAND Flash Memory Lifetime [note 1]

**DOI:** 10.3390/mi15121447

**Published:** 2024-11-29

**Authors:** Jialin Wang, Yi Fan, Yajuan Du, Siyi Huang, Yu Wan

**Affiliations:** 1College of Electrical Engineering, Naval University of Engineering, Wuhan 430033, China; 1310041031@nue.edu.cn; 2School of Computer and Artificial Intelligence, Wuhan University of Technology, Wuhan 430070, China; fy2685018622@whut.edu.cn (Y.F.); huangsiyi@whut.edu.cn (S.H.); wanyu@whut.edu.cn (Y.W.)

**Keywords:** 3D NAND’s flash memory, layer endurance variation, SSD lifetime, bit error rate

## Abstract

With vertical stacking, 3D NAND’s flash memory can achieve continuous capacity growth. However, the endurance variation between the stacked layers becomes more and more significant due to process variation, which will lead to the underutilization of many pages and seriously affect the lifetime of 3D NAND’s flash memory. We investigated the endurance variation characteristics between layers and divided the stacked layers into the top, middle, and bottom layers according to the endurance characteristics. We found that the endurance of the bottom layer pages is much weaker than that of the other two layers, which is the primary factor that affects the lifetime of 3D NAND’s flash memory. In response to this endurance variation feature, we proposed a new layer-aware write strategy, called LA-Write. First of all, the write–skip unit in LA-Write will reduce the wear pressure of the pages through write–skip operations. Secondly, LA-Write maintains a layer-aware table, which stores the probability of pages in different layers performing the write–skip operation. Setting the probability of the bottom pages to the highest value will result in more write–skip operations on the bottom layers, mitigating endurance variations between layers. We carried out our experiments of LA-Write on DiskSim, a popular SSD simulator. Compared to existing schemes, experimental results show that LA-Write can greatly increase SSD’s lifetime.

## 1. Introduction

With the rapid development of artificial intelligence, cloud computing, and Internet of Things technologies, the scale of global data is experiencing explosive growth [1]. To meet the demand for larger storage capacity, several non-volatile storage media such as Resistive RAM (RRAM) [2,3], Magnetic RAM (MRAM) [4], and flash memory are used. Compared to other emerging memories, flash memory remains the most viable option for cost-sensitive, high-capacity applications. Flash memory suppliers have been actively increasing the number of vertical layers to adapt to this demand [5]. As a result, the structure of flash memory has gradually shifted from 2D flash memory to 3D flash memory. However, compared with 2D flash memory, the manufacturing process of 3D flash memory is more complex. Specifically, because 3D flash memory is manufactured using a vertical continuous etching process from the top layers to the bottom layers, there are huge process variations between stacked layers [6,7,8]. The process variations cause flash pages within a flash block to wear at different rates. Since the reliability strategy of flash memory is such that once any page in a block exceeds the ECC error correction capability, the entire block will be considered a bad block [9], and the Flash Translation Layer (FTL) will stop using the bad block. However, pages with better endurance within blocks are underutilized. Variations in endurance between layers will result in prematurely worn pages in the block, which will cause the flash memory to lose available blocks faster, making the flash die faster.

Recently, some researchers have proposed solutions to the flash memory lifetime problem caused by the endurance variation between layers [10,11,12]. Wei et al. proposed a Bad Page Management (BPM) scheme [11]. When the bit error rate (BER) of a page in a block exceeds the error correction capability of ECC, BPM will not immediately mark the entire block as a bad block, and FTL will continue to use the block until the number of bad pages in the block exceeds the threshold preset by BPM. Experimental results show that BPM can extend the lifetime of flash memory. Jimenez et al. define pages with a faster BER growth rate as weak pages. When writing to a block, they avoid writing to weak pages to relieve the write pressure of weak pages and prevent weak pages from exceeding the error-correction capability of ECC too early, thereby effectively extending the lifetime of the block. Hong et al. proposed the GuardedErase scheme [12], which extends the life of the block by reducing the page erase pressure.

To quantify the endurance variation between stacked layers caused by NAND’s process variations, we conducted process characterization experiments. Based on the endurance characteristics obtained from the experimental results, we divided the stacked layers into top, middle, and bottom layers. We observed significant inter-layer BER variations in the flash blocks tested. Specifically, the bottom layer pages had the fastest growth in BER, much faster than the other two layers, followed by the top layer pages. The middle layer pages grow the slowest and have the best endurance.

Based on the above observations, we propose a new layer-aware write strategy, called LA-Write, to balance the endurance variations among layers to extend the lifetime of the flash-based Solid State Disk (SSD) with the smallest possible performance overhead. On the one hand, LA-Write uses the write–skip operation to skip the current page, improving the endurance of the skipped page. On the other hand, LA-Write maintains a table called the layer-aware table, which stores the probability of performing write–skip operations on pages of different layers. Specifically, we assigned the highest probability to the bottom layers with the least endurance, the top layers second, and the middle layers the lowest. Through the above two crucial implementations, some of the pressure of write requests will be transferred to the middle and top layers with relatively better endurance, balancing the endurance of the three layers and extending the lifetime of the flash memory.

In order to evaluate LA-Write, we conducted comprehensive experiments with real workloads on the Disksim simulator [13], and the results showed that LA-Write can improve the SSD lifetime by 31% on average. The main contributions of this work are as follows.

We preliminarily investigate the characteristics of inter-layer error variation and intra-layer error variation. Our findings indicate that intra-layer BER exhibits a high degree of similarity, whereas inter-layer BER demonstrates significant variation.Based on the inter-layer endurance characteristics, we proposed LA-Write and introduced in detail two pivotal implementations of LA-Write: a layer-aware table and write–skip unit.We conducted a rigorous evaluation of LA-Write and showed that LA-Write can significantly improve SSD’s lifetime with very little performance overhead.

The rest of this paper is organized as follows. Section 2 introduces the layer-stacked structure of 3D flash memories and the reliability management strategy of 3D NAND’s flash memory. Also, we introduce our preliminary research and observations on the BER variation of 3D NAND’s flash memory in this section. Section 3 describes the specific design of LA-Write in detail. Section 4 gives the experimental setup and experimental results of LA-Write and analyzes the experimental results in detail. Section 5 concludes this paper.

## 2. Background and Motivation

In this section, we first review the layer-stacked structure of 3D flash memories. Next, we discuss reliability management in 3D NAND’s flash memory and our motivation.

### 2.1. Structure of 3D NAND’s Flash Memory

Figure 1 shows a part of the 3D flash memories, where all the layers marked in green along the z-axis form a flash block [14]. Although the capacity of the flash block is substantially raised by vertically stacking multiple layers [15,16], due to process variation and etching technologies [17,18], the diameters of the cylindrical channels connecting the stacked layers vertically vary significantly. As a result, manufacturers have difficulty in manufacturing identical stacks of 3D flash layers, and this characteristic of 3D flash memories leads to significant variations in endurance between pages in different layers.

### 2.2. Reliability Management in NAND Flash Memory

Due to the out-of-place update characteristics [19] of NAND flash memory, the flash memory block needs to be erased each time before programming with new data. Each pair of erase and program operations is called a P/E cycle [20]. However, after repeated program/erase operations are performed on the block, the amount of charge trapped in the tunnel oxide layer increases. The trapped charges make it difficult for the NAND cells’ threshold voltage levels to remain within their expected voltage range, which significantly affects the reliability of the NAND cells [10,21]. In other words, flash blocks can only go through a limited number of P/E cycles before they wear out. As the P/E cycle increases, the BER of the data in the block increases gradually [22], resulting in errors in the data stored in the cells.

For reliability reasons, when the BER of a page exceeds the error correction capability of the ECC engine, manufacturers believe that the P/E cycles that the page has experienced at this time are the maximum P/E cycles that the page can experience, that is, the NAND’s endurance [23]. Due to the process variation of 3D NAND, the endurance of each layer of pages is different, and the maximum P/E cycles it can experience are also different. However, when the BER of any page in the block exceeds the ECC error correction capability, the entire block will be considered a bad block [24]. In other words, the number of maximum allowed P/E cycles of a block depends on the P/E cycles that the least endurable page in the block can experience; there are inclined to be many pages in a bad block that have experienced P/E cycles that are far from the maximum P/E cycles they can experience, which leaves the pages with good endurance in the block underutilized. Compared with the ideal 3D NAND’s flash memory with balanced endurance of each layer, the current 3D NAND’s flash memory will reach the end of its lifetime faster.

### 2.3. Motivation

To explore potential avenues to extend the lifetime of 3D NAND’s flash memory and quantify the variation in page BER between different layers, we conduct two experiments with inter-layer BER and intra-layer BER for 3D MLC flash with 32 stacked layers and analyze the results. The experiments are based on the SSDmodel module in the DiskSim simulator.

Inter-Layer Error Variation: The variation of page BER in different layers is illustrated in Figure 2. Among them, the curves with different colors indicate the number of disparate P/E cycles experienced. Observing Figure 2, we can find that the BER values of the pages located in each layer show a significant increasing trend with the increase in P/E cycles, especially from the 2nd to the 15th layers; the BER is significantly higher than other layers. According to the BER’s growth rate characteristics, we divide the 32-layer stacking layer into three layers: bottom layers (L2–L15), top layers (L29–L32), and middle layers (L1, L26–L28). Among them, the order of page endurance from weak to strong is the bottom, top, and middle layers.

Intra-Layer Error Variation: As shown in Figure 2, the solid and dashed lines of the same color indicate the upper and lower pages of the same flash layer, respectively. It is not difficult to find out from the experimental results that, unlike the inter-layer error variation, the BER variation trends of the upper and lower pages within the layer are strikingly similar. The etching environment of the same layer is the same, and theoretically, the pages within the layer should have a similar endurance, which is consistent with the results obtained from our experiments.

In conclusion, we have found through the above experiments that the main reason restricting SSD lifetime is that the BER growth rate of the pages of the bottom layers is much higher than that of the other two layers, resulting in flash blocks being more rapidly regarded as bad blocks by FTL. Our proposed method will also work from this perspective to balance the endurance of the three layers and specifically to alleviate the wear pressure on the bottom pages, which will be elaborated on in detail in the next section.

## 3. Materials and Methods

In this section, we first outline the overall architecture of LA-Write. Then, we detail two important implementations of the LA-Write method. Finally, we analyze the overhead of the proposed method.

### 3.1. Architectural Overview of LA-Write

Figure 3 shows the comprehensive architecture of LA-Write, among which it is worth paying attention to the layer-aware table and the write–skip unit proposed by us. When FTL selects the active page [25] as the target for incoming write requests, the traditional write strategy follows FTL’s selection and directs the write request to the active page selected by FTL. In contrast, LA-Write may change the address of the write request as determined by FTL. In the SSD Controller with the LA-Write mechanism, first, determine which layer the active page selected by FTL belongs to, then query the layer-aware table according to the layer to obtain the probability, and finally, the write–skip unit will obtain the final active page of the write request based on the probability. The purpose of LA-Write is to direct more write operations to more endurable layers, relieve the burden of less endurable layers, and balance the endurance of pages between layers.

### 3.2. Design and Implementation of LA-Write

#### 3.2.1. Layer-Aware Table

Since the BER of the bottom layers grows much faster than that of the other layers, when the BER of the pages of the bottom layers exceeds the ECC’s error-correction capability, the BER of the pages of the middle and top layers is still far from the upper limit of the ECC’s error-correction capability, and this extreme imbalance in inter-layer endurance will seriously affect the lifetime of the entire block. Therefore, we designed the layer-aware table intending to mitigate the bottom layers’ BER growth rate. The layer-aware table stores the probability of executing a write–skip operation on the pages of each layer, where the write–skip operation causes the write operation to skip the current page and move the active page backward by one page in sequence. The specific probability settings are shown in Table 1. The write–skip unit will decide whether to perform the write–skip based on the probability in this table. That is, as long as the probability of performing the write–skip operation is set relatively high for the bottom layers, relatively low for the middle layers, and in-between for the top layers, the number of write–skip operations performed on the bottom layers will become correspondingly higher, and more write operations will be guided to more endurable middle and top layers, which would balance the endurance between layers and prolong the lifetime of the SSD.

The probability of each layer in the layer-aware table cannot be set too high or too low. If the probability is set too high, there will be too many write–skip operations in the processing of write requests, which will reduce the available space in flash memory, causing garbage collection and exacerbating flash block wear pressure. If the probability is set too low, the bottom layers pages will trigger write–skip fewer times, and the endurance variation between the bottom and other layers cannot be alleviated, nor can the growth rate of the bottom layers’ BER be effectively mitigated. We believe that the 20%, 10%, and 5% probabilities corresponding to the bottom, top, and middle layers, respectively, are best suited to balancing the endurance between different layers in the block, and the reasons why the probability values are set this way are described in detail in the next section.

#### 3.2.2. Write–Skip Operation and Workflow of LA-Write

The wear of flash memory is mainly caused by repeated write and erase operations, so it is possible to reduce the wear of pages by reducing the writing of pages with poor endurance to extend the lifetime of flash memory. The write–skip unit in LA-Write was born for this purpose. The write–skip unit uses the probability from the layer-aware table to determine whether to execute a write–skip operation. Specifically, the write–skip operation causes the target address of the write request to skip the current active page and causes both the target address and the active page to move back one page in sequence. The write–skip unit will repeat this process until it chooses not to perform write–skip. At this point, the target address of the write request is determined, which is the current active page. From the first time the layer-aware table passes the probability to the write–skip unit determining the final target address of the write request, this entire process is called the WS Process.

The workflow of LA-Write can be divided into two steps. First, LA-Write obtains the probability of performing the write–skip operation on the active page allocated by FTL. Second, the write–skip unit in LA-Write executes the WS Process to obtain the final target address. The specific process is as follows.

LA-Write will first determine which of the three layers of the active page determined by FTL for this write request belongs to and then look up the probability corresponding to this layer in the layer-aware table. After performing the above steps, LA-Write obtains the probability that the WS Process executes write–skip for the first time.During the execution of the WS Process, if the write–skip unit decides to perform the write–skip operation this time, then after executing the write–skip this time, the final target address still cannot be determined, and LA-Write moves the active page to the next page of the current active page in order. After that, LA-Write determines which layer the currently active page belongs to, then searches the layer-aware table for the corresponding probability according to this layer and determines whether to execute write–skip this time based on this probability. The above process will be repeated until LA-Write decides not to perform the write–skip at some point. At this time, LA-Write has determined the final target address, which is the current active page, and LA-Write returns the current active page to FTL.

LA-Write probabilistically performs write–skip on pages to reduce page wear pressure. Different layers have corresponding probabilities that determine the number of write–skip operations performed on their pages, resulting in varying degrees of relief of the wear pressure of each layer. In LA-Write, it is quite possible to relocate the target page with poor endurance assigned by FTL to a more endurable page, which will reduce the wear of the poor-endurance layer, narrow the lifetime variation between different layers, and ultimately extend the lifetime of SSD.

### 3.3. Overhead Analysis

Time overhead of executing LA-Write: Since the layer-aware table is stored in DRAM in the flash controller, the table lookup time overhead is negligible since the time to perform a read to DRAM is several orders of magnitude shorter compared to a single write to NAND flash.

Extra Space Overhead for LA-Write: We use the 64 GB flash with 32 layers in the experiment. The probability of each layer takes only 1 Byte. Thus, the overall storage overhead is 32 Bytes, which is negligible.

## 4. Results and Discussion

In this section, we will introduce the experimental platform, experimental settings, experimental configurations, and comparative experimental settings; analyze the experimental results; and evaluate the effectiveness of our proposed LA-Write in extending SSD lifetime.

### 4.1. Experimental Setup

To evaluate the effectiveness of the proposed technique, we implemented LA-Write on Disksim [26], a well-known disk simulator whose extensibility allows for precise modeling of unique 3D NAND characteristics, such as complex multi-layer architectures and specific latency behaviors across layers. Although we use DiskSim to simulate 3D NAND, we believe LA-Write can work on real devices.

We used a Disksim with SSD extensions to simulate 64 GB 3D MLC flash. There are four channels in the flash memory, each channel has two flash chips, each chip has eight planes, each plane has 2048 blocks, each block contains 384 pages, and the page size is 16 KB. Other experimental settings are shown in Table 2, which are mainly based on the specifications of the 3D NAND chip manufactured by Samsung [27]. It is worth mentioning that the overprovisioning ratio is set at 15%, in line with most commercial SSDs. To effectively utilize the overprovisioning area, GC is invoked when the number of free blocks is less than 5% of the total number of blocks.

We selected seven data-intensive storage workloads from the MSR Cambridge trace and UMass Trace Repository, which were stg1, prn0, proj1, w0, ts0, prn1, and proxy0, as shown in the Table 3. In order to ensure that the results of the experiment were broad and general, the workloads used for the experiment were carefully selected, covering the workload dominated by read and write requests, as well as the workload with a balanced ratio of read and write. This choice ensures that the performance of the simulator in different scenarios is fully considered, and the effect of the write–skip mechanism on prolonging 3D NAND flash lifetime can be more comprehensively evaluated. We use LA-Write to compare with three existing schemes: Baseline, BPM [11], and GuardedErase [12].

(1) Baseline: The Baseline represents traditional sequential writing. When FTL selects a free block for page allocation, it allocates all pages in the block in sequence. Only when all pages in the current block are allocated will it select another free block to allocate pages.

(2) BPM: When the BER of a page in a block exceeds the ECC error correction capability, BPM will not immediately mark the entire block as a bad block. Instead, BPM will only mark the page as a bad page and continue to use the block normally. When the number of bad pages in a block exceeds the preset threshold, the entire block will be marked as a bad block and the block will stop being used.

(3) GuardedErase: The GuardedErase scheme slows down the growth of page BER by using low voltage to erase pages. For example, assuming that the effective erase voltage of a block is 17 V, GuardedErase will adjust the erase voltage of some pages in the block to 14 V. The effective erase voltage of these pages is reduced by 3 V, which reduces the erase pressure they are subjected to and slows down the growth of BER. However, to implement this scheme, multiple linked lists must be maintained for each flash memory block. Each time a low voltage erase is performed on some of the pages in the block, the linked list of the block needs to be searched. In addition, in order to maintain the accuracy of the linked list, the linked list of the block needs to be updated every 100 P/E cycles. The maintenance and update of the linked list will cause non-negligible overhead.

### 4.2. Lifetime Improvement

The SSD lifetime is quantified by the total amount of data written when the SSD dies [11,12,28]. It is worth noting that the total amount of data written is not equivalent to the total amount of data ultimately stored in the flash memory but refers to all the data written from the time the flash memory starts to be used to the time when the number of bad blocks in the flash memory reaches a certain threshold and there are no available reserved blocks to replace the bad blocks (i.e., the flash memory dies). Therefore, in order to evaluate the effect of the LA-Write on extending the lifetime of the 3D NAND’s flash memory, we will also use the total amount of data written when the flash memory dies as the measure of the lifetime of the flash memory. For convenience, the experimental results were normalized. As shown in Figure 4, it is not difficult to find that LA-Write significantly extends the lifetime of flash memory. Specifically, compared with the Baseline, the lifetime is extended by an average of 31%. Compared with the BPM scheme and GuardedErase scheme, the experimental results also show superiority, extending the lifetime by 10% and 5%, respectively, on average. Compared with BPM, why is the improvement of LA-Write over GuardedErase in extending the lifetime of flash memory less obvious? This is for two reasons (1) BPM prolongs the lifetime of the flash memory by continuing to use the flash block after it has become a bad block. However, the experiments in Section 2.3 show that the BER of the bottom layers is greater than that of the middle layers and the top layers, and the BER of pages in the same layer shows similarity. This means that when a bottom page of a block exceeds BER, other pages at the bottom layer of the block will also exceed the error correction capability of ECC, which will cause the number of bad pages in the block to reach the preset threshold very quickly. BPM is not ideal for extending the lifetime of flash memory. (2) GuardedErase reduces the erase voltage of pages with larger BER to slow down the BER growth of these pages. This scheme effectively delays the time when the block becomes a bad block, so GuardedErase performs better than BPM in extending the lifetime. However, since the effective erase voltage is only reduced to 3 V, the effect is limited. Since LA-Write extends the lifetime of the bottom layers pages that play a key role in determining the lifetime of the entire block, LA-Write obtains the best experimental results.

In order to more intuitively verify the effect of the write–skip mechanism proposed in this paper on extending the life of flash memory, we selected some time nodes during the use of the SSD and counted the amount of written data at the corresponding time nodes. The results are shown in Figure 5. When the amount of written data does not change with time, this is the time node when the SSD dies, which we marked with a red dotted line. It can be seen that the time when the SSD using each strategy dies is consistent with the results of Figure 4. Since the Baseline does not use any strategy to extend the lifetime, it dies first. Since the strategy used by GuardedErase is more effective than BPM in extending the lifetime, BPM dies before GuardedErase. Finally, LA-Write dies after the SSDs of the other three schemes die.

### 4.3. Page Utilization

The experiment in Section 2.3 shows that when the BER of the bottom layer pages exceeds the error correction capability of ECC and the entire block is marked as a bad block and no longer used, the top layer pages and the middle layer pages are not fully utilized. LA-Write uses the Layer-Aware Table to perform write–skip operations on the bottom-layer pages more frequently, and the corresponding write operations will be allocated to the middle and top layer pages, which fully utilize the middle and top layer pages. In order to verify whether the middle and top layer pages are fully utilized, we count the BER of all layers in the bad block, and the experimental results are shown in Figure 6. We can find out the following.

The BER value at the highest point in the figure represents the error correction capability of the ECC. When the BER value of any page in the block exceeds the error-correction capability of the ECC, the entire block will be marked as a bad block. Therefore, it can be reversely inferred that the BER value at the highest point is the error correction capability of ECC. The greater the height difference between a point and the highest point, the further the BER of the point exceeds the error correction capability of the ECC, which means that it is not fully utilized.Under the Baseline, when the BER of the bottom layer page exceeds the ECC error correction capability and the entire flash block is marked as a bad block and no longer used, the BER of the top and middle layer pages is far from reaching the ECC error-correction capability, which means that these pages are not fully used. This is consistent with the conclusions of the preliminary experiments in Section 2.3.The BPM scheme and the GuardedErase scheme cannot fully utilize the middle and top layer pages. Since they do not change the way the flash memory is written, they still use the traditional sequential write method, just like the Baseline. Therefore, although the block can undergo more P/E cycles, the BER of all layers will increase, and the overall BER curve trend will be still similar to Baesline, which means that the middle- and top-layer pages will still not be fully utilized.LA-Write can make full use of the middle- and top-layer pages. It can be seen from Figure 6 that the BER curve of LA-Write is very different from the other three schemes. The number of points with lower BER values in LA-Write is significantly less than that in the other three schemes, and the BER of all layers is relatively balanced, which shows that the middle and top layer pages are fully utilized.

### 4.4. Performance Overhead

IOPS (input/output operations per second) is commonly used to measure the performance of flash memory. IOPS represents the number of read/write operations that a flash memory is capable of performing in one second, and a higher IOPS indicates that the flash memory has a faster response and a higher performance. In order to explore the impact of the LA-Write mechanism on read and write performance, we count the IOPS in flash memory using Baseline and LA-Write, respectively. We found that LA-Write had minimal impact on performance, as shown in Figure 7. Specifically, the average read and write performance decreased by only 3%. Among these workloads, workload proxy0 experienced the most performance degradation. The main reason is that the higher write request ratio in proxy0 makes it easier to trigger multiple write–skip operations, which has a certain impact on the performance of write operations. Overall, LA-Write achieves a significant extension of flash memory lifetime with almost negligible performance overhead.

### 4.5. Sensitivity Study

In this section, we analyze the impact of probability in the layer-aware table on extending the lifetime of flash memory. In the experiments in this section, the measurement indicators for the flash memory lifetime are the same as those in the previous experiments, that is, the total amount of written data when the flash memory dies is counted, and the results are normalized based on the Baseline. Among them, LA-Write (20, 10, 5) means that the probabilities of executing write–skip on the bottom-, top-, and middle-layer pages are 20%, 10%, and 5% respectively. We conducted experimental analysis on the cases where the probability groups in the layer-aware table are small and large. The results are as follows.

(1) When the bottom-, top-, and middle-layer probabilities are set relatively small:

The layer variation error characteristics obtained through preliminary experiments in this paper show that the BER growth rate of the bottom layer is much faster than that of the top and middle layers. In this case, even if the probabilities of the bottom, top, and middle layers are set to be smaller, the probability of the bottom layer should be greater than the probabilities of the top and middle layers. Therefore, the experiment selected LA-Write (15, 5, 3), LA-Write (15, 10, 5) and LA-Write (20, 10, 5) and compared their effects on extending the lifetime of the 3D NAND’s flash memory. The experimental results are shown in Figure 8.

We can find that LA-Write (20, 10, 5) has the best effect in extending the lifetime. This is because the other two groups set the probability of the bottom layers to be lower, which results in too few write–skip executions on the bottom pages, and the effect of slowing down the growth of the bottom layer pages BER is limited, which also affects the effect of LA-Write in extending the lifetime of the flash memory.

(2) When the bottom-, top-, and middle-layer probabilities are set relatively large:

In order to more comprehensively explore the effect of each layer probability setting in LA-Write on extending the lifetime, we conducted two sets of comparative experiments. First, we considered the case where only the probability of the bottom layer is large. We selected LA-Write (20, 10, 5), LA-Write (30, 10, 5), and LA-Write (40, 10, 5) for comparison. The results are shown in Figure 8b. In addition, we also considered the case when the probabilities of the bottom layer, top layer, and middle layer are relatively large. We selected LA-Write (20, 10, 5), LA-Write (20, 15, 10), and LA-Write (30, 15, 10) and compared their effects on extending the lifetime of the flash memory. The results are shown in Figure 8c.

By combining the probability settings of the two groups of experiments, it can be seen that LA-Write (20, 10, 5) has achieved the best effect in extending the life of the flash memory. Compared with LA-Write (20, 10, 5), whether only the probability of the bottom layer is set larger or the probability of all three layers is set larger, the effect of LA-Write in extending the lifetime of the flash memory will be worse. This is because when the probability becomes larger, the available write space of the flash memory will decrease. When the available write space of the flash memory is small, if the write request is frequent, the flash memory can only perform garbage collection more frequently and erase the block in order to continue writing to the block. However, the erase operation will increase the BER of the pages in the block, thereby accelerating the death rate of the block. In this case, although LA-Write can slow down the growth rate of the BER of the pages in the block, frequent erase operations will make LA-Write’s effect of extending the lifetime of the flash memory worse.

In summary, this section comprehensively considers the setting of the probability in the layer-aware table and conducts sensitivity experiments. According to the experimental results, the LA-Write has the best effect in extending the lifetime of the flash memory when the probabilities of the bottom layer, top layer, and middle layer are set to 20%, 10%, and 5%, respectively.

## 5. Conclusions

This article proposes LA-Write, which effectively extends the lifetime of 3D NAND flash by directing more write requests from the least endurable bottom layers to the more endurable middle and top layers. LA-Write extends the lifetime of flash memory with two key implementations: (1) write–skip unit: by performing write–skip operation on the page, the lifetime of the page that performs the write–skip operation is extended; (2) layer-aware table: according to the results of motivation experiment, a reasonable probability is set for each layer to perform the write–skip operation, making it more likely that the bottom layers will perform write–skip operations. Our results on the DiskSim simulator show that by performing the write–skip operation on the page according to the probability in the layer-aware table, LA-Write achieves a significant extension of the lifetime of 3D NAND flash with negligible performance overhead.

## Figures and Tables

**Figure 1 micromachines-15-01447-f001:**
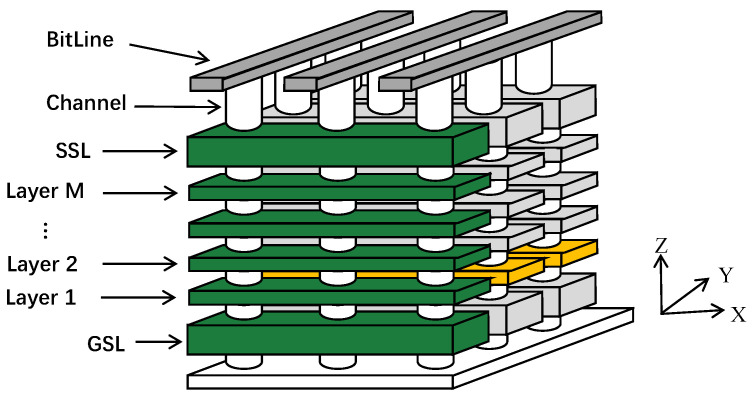
Overview of 3D flash memory. The orange highlighted flash memories are on the same layer, and the green highlighted flash memories form a flash block.

**Figure 2 micromachines-15-01447-f002:**
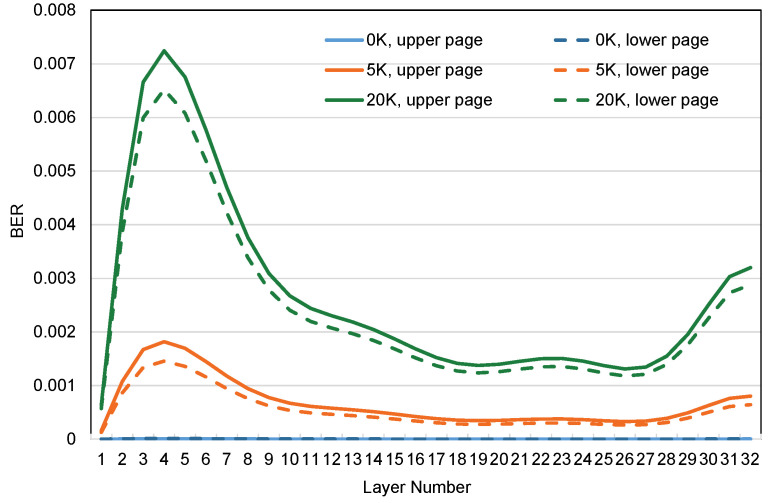
BER variations of pages in different layers.

**Figure 3 micromachines-15-01447-f003:**
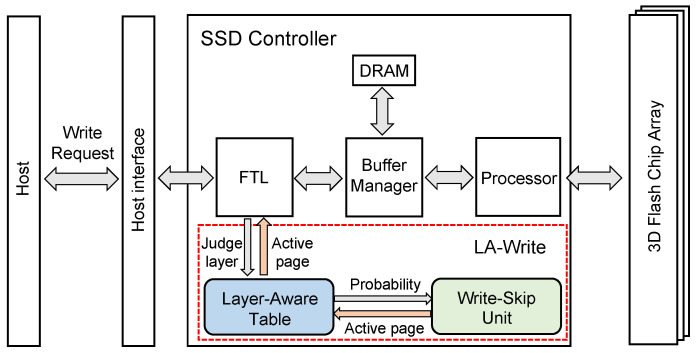
The architecture overview of LA-Write. Two components are added to SSD to implement LA write: (1) Layer-Aware Table intends to mitigate the bottom layers’ BER growth rate and (2) Write-Skip Unit is used to determine whether to exucute a write-skip operation.

**Figure 4 micromachines-15-01447-f004:**
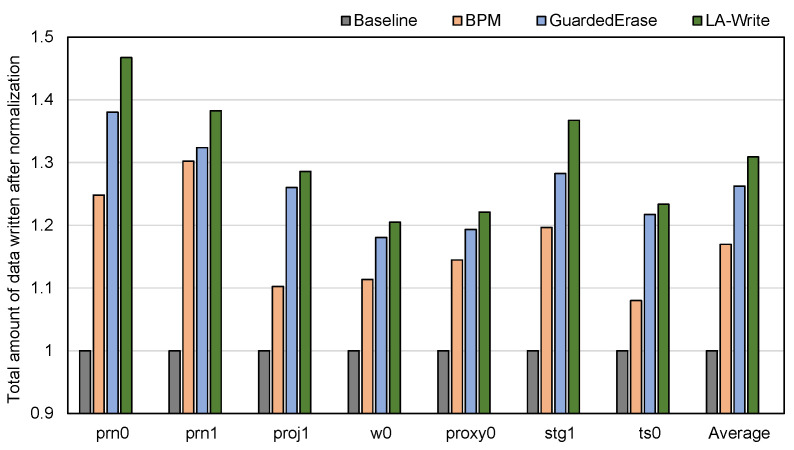
The amount of data written under different scenarios.

**Figure 5 micromachines-15-01447-f005:**
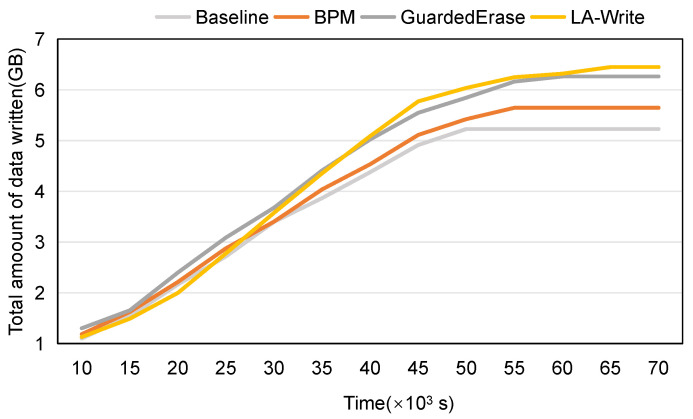
The amount of written data as time goes by.

**Figure 6 micromachines-15-01447-f006:**
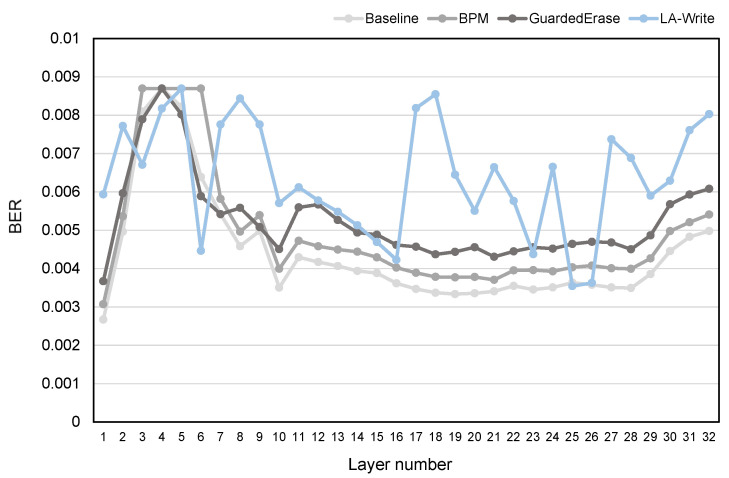
BER of all layers in a bad block.

**Figure 7 micromachines-15-01447-f007:**
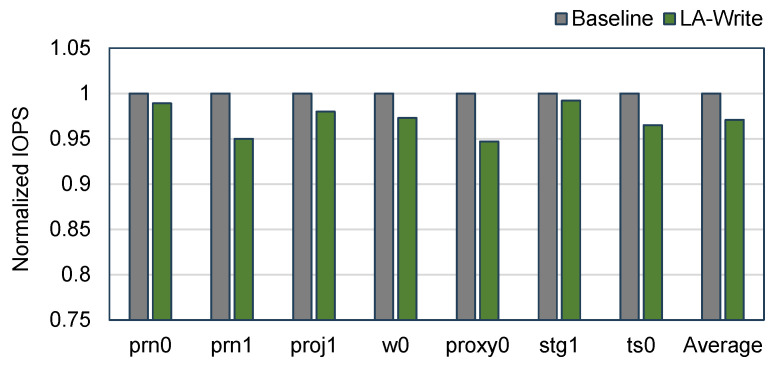
Comparison of performance overhead.

**Figure 8 micromachines-15-01447-f008:**
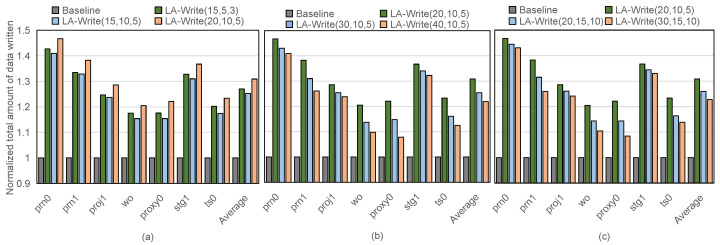
The effect of LA-Write on extending the lifetime of 3D NAND’s flash memory (**a**) when the probability of three layers is relatively small; (**b**) when the bottom probability is large; (**c**) the probability of three layers is relatively large.

**Table 1 micromachines-15-01447-t001:** Layer-aware probability among different layers.

	Bottom Layers	Top Layers	Middle Layers
Layer Number	L2–L15	L29–L32	L1, L16–L28
Probability	20%	10%	5%

**Table 2 micromachines-15-01447-t002:** Experimental setups.

Item	Specification
Flash size (GB)	64
Page size (KB)	16
Number of blocks per plane	2048
Number of pages per block	384
Page write latency (μs)	600
Page read latency (μs)	49
Data transfer rate (Mbps)	553
Page program time (ms)	0.7
Block erase time (ms)	4
Clean block threshold (%)	5
Overprovisioning area (%)	15

**Table 3 micromachines-15-01447-t003:** Details of workloads used in the experiments.

Trace	Read Requests Number	Write Requests Number	Write Request Ratio
stg1	1,132,587	301,067	0.21
prn0	710,297	289,703	0.29
proj1	1,006,576	1,320,260	0.55
w0	3,848,447	14,477,493	0.79
ts0	2,896,685	15,207,594	0.84
prn1	208,094	1,339,720	0.86
proxy0	81,885	1,555,824	0.95

## Data Availability

The data presented in this study are available on request from the corresponding author. The data are not publicly available due to confidentiality request.

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
