# Peer review of "Balancing Page Endurance Variation Between Layers to Extend 3D NAND Flash Memory Lifetimeâ€"

_micromachines, 2024, doi:10.3390/mi15121447_

Round 1
Reviewer 1 Report
Comments and Suggestions for Authors
The authors proposed LA-Write strategy to extend the lifetime of 3D NAND flash with key implementations of Write-Skip Unit and Layer-Aware Table, and simulated on DiskSim. More issues should be noted after a major revision as the following suggestions:
1. The full name should be written for the first occurrence of professional terms in the text, such as SSD, LA, BER etc.
2. The research article structure usually includes an introduction, materials and experiment, results and discussion, and a conclusion. In this article, part 2, the background and part 1, the introduction are duplicated. Also, part 3, layer error variation of 3D NAND flash memory should be part of the experiment results. Please adjust the content distribution and focus on the novelty and result of the work.
3. The numerical values on the x-axis are not displayed in Figure 6.
4. The method part is too long, the authors can list and summarize the parameters, and describe the experimental process in points. The same type of results, such as Figure 9-11 can be combined into a large image and annotated with (a), (b), and (c) for analysis.
5. The authors stated that the process variation between stacked layers seriously affects the lifetime of 3D NAND flash memory. More explanations should be added. Also, to get the most out of LA-Write, what are the requirements or improvements in the fabrication process?
6. In addition to the DiskSim simulator, are there any results verified on the device?
Author Response
Comment 1: The full name should be written for the first occurrence of professional terms in the text, such as SSD, LA, BER etc.
Response 1: The full name of professional terms (SSD, LA, BER, FTL and so on) have been written when they first occur. Please refer to Section 1.
Comment 2: The research article structure usually includes an introduction, materials and experiment, results and discussion, and a conclusion. In this article, part 2, the background and part 1, the introduction are duplicated. Also, part 3, layer error variation of 3D NAND flash memory should be part of the experiment results. Please adjust the content distribution and focus on the novelty and result of the work.
Response 2: We have adjusted the structure of manuscript to make it more concise. Because the background is important, we have placed the original section 3 as the motivation in the second chapter, and we have retained section 2. Please refer to our revised manuscript.
Comment 3: The numerical values on the x-axis are not displayed in Figure 6.
Response 3: We have supplemented on Figure 6. Please refer to our revised manuscript.
Comment 4: The method part is too long, the authors can list and summarize the parameters, and describe the experimental process in points. The same type of results, such as Figure 9-11 can be combined into a large image and annotated with (a), (b), and (c) for analysis.
Response 4: We have modified the redundant part of the method section to make this part more concise. We have also merged Figures 9-11. Please refer to our revised manuscript.
Comment 5: The authors stated that the process variation between stacked layers seriously affects the lifetime of 3D NAND flash memory. More explanations should be added. Also, to get the most out of LA-Write, what are the requirements or improvements in the fabrication process?
Response 5: in section 2.3, we have explained that the main reason restricting SSD lifetime is that the BER growth rate of the pages of the bottom layers is much higher than that of the other two layers, resulting in flash blocks being regarded as bad blocks by FTL more rapidly. To use LA-write, SSD manufacturers only need to modify the logic of the SSD controller, without modifying the hardware. Please refer to our revised manuscript.
Comment 6: In addition to the DiskSim simulator, are there any results verified on the device?
Response 6:
we only implemented our method on DiskSim simulator. Our work is not a theoretical, but a practical suggestion. And the overhead has been evaluated in section 3.3. Because we only propose improvement methods at the software layer, it is feasible to apply this method to formal devices.

Reviewer 2 Report
Comments and Suggestions for Authors
The authors proposed LA-Write method, which extends the life time of NAND Flash memory. LA-Write method had larger lifetime than the other ones. In addition, the authors showed various data related to the characteristic of NAND flash memory. This information is helpful for the readers. However, there are some concerns about the introduction and the results. If the authors appropriately revise the manuscript, this study will meet the criteria for the publication in Micromachines.
Comment 1: In Figure 2, BER seems to oscillate periodically. What happens in this layer number dependence of BER?
Comment 2: In Figure 6, total amount of data written in LA-Wire and GuardedErase crossed at the certain time. What happens?
Comment 3: There are less competitive studies in the beginning introduction. To attract much attention of the readers, the authors should additionally comment on other memories. For example, nanostructured memory devices are also attracting much attention because of their high performance: RRAM (Sci. Technol. Adv. Mater. 21, 195 (2020).), MRAM (Sci. Rep. 7, 16729 (2017).), etc. The authors should revise the manuscript and cite the related references.
Author Response
Comment 1: In Figure 2, BER seems to oscillate periodically. What happens in this layer number dependence of BER?
Response 1:
sorry, I don't understand what you mean by “oscillate periodically”. As the proram/erase (P/E) cycle increases, the BER of flash will also increase. Note that the x-axis in Figure 2 represents the number of flash layer rather than time, we divide the 32-layer stacking layer into three layers: bottom layers (L2-L15), top layers (L29-L32), and middle layers (L1, L26-28). And the important message conveyed in this figure is that due to differences in manufacturing processes, the BER of the middle layer is higher than that of other layers. For better understanding, we have merged this section into section 2. Please refer to section 2.3 of our revised manuscripts.
Comment 2: In Figure 6, total amount of data written in LA-Wire and GuardedErase crossed at the certain time. What happens?
Response 2: Since the LA-Write strategy will have a slight impact on the read and write performance, the LA-Write and GuardedErase curves may cross. The cross of the curves only means that the average write rates of LA-Write and GuardedErase are equal from the beginning to the cross of the curves.
Comment 3: There are less competitive studies in the beginning introduction. To attract much attention of the readers, the authors should additionally comment on other memories. For example, nanostructured memory devices are also attracting much attention because of their high performance: RRAM (Sci. Technol. Adv. Mater. 21, 195 (2020).), MRAM (Sci. Rep. 7, 16729 (2017).), etc. The authors should revise the manuscript and cite the related references.
Response 3: two high performance memories(PRAM and MRAM) have been mentioned and cited in the beginning of Section 1, and the related references([2], [3]) has been cited. Please refer to our revised manuscript.

Round 2
Reviewer 1 Report
Comments and Suggestions for Authors
The authors have made modifications according to the review report, and the manuscript could be accepted in present form.
Author Response
OK, Thank you for your comments.
Reviewer 2 Report
Comments and Suggestions for Authors
The paper (Sci. Technol. Adv. Mater. 21, 195 (2020).) is missing although the authors told me that the manuscript was revised. Please reconfirm this. If the authors revise the manuscript, this study will be accepted.
Author Response
Comment 1: The paper (Sci. Technol. Adv. Mater. 21, 195 (2020).) is missing although the authors told me that the manuscript was revised. Please reconfirm this. If the authors revise the manuscript, this study will be accepted.
Response: We have added the reference in section 1. Please refer to our revised manuscript.
